# Using the Non-Adoption, Abandonment, Scale-Up, Spread, and Sustainability (NASSS) Framework to Identify Barriers and Facilitators for the Implementation of Digital Twins in Cardiovascular Medicine

**DOI:** 10.3390/s23146333

**Published:** 2023-07-12

**Authors:** Peter D. Winter, Timothy J. A. Chico

**Affiliations:** 1School of Sociology, Politics, and International Studies (SPAIS), University of Bristol, Bristol BS8 1TU, UK; 2Department of Infection, Immunity and Cardiovascular Disease (IICD), University of Sheffield, Sheffield S10 2RX, UK; t.j.chico@sheffield.ac.uk

**Keywords:** medical digital twins, cardiovascular diseases, personalised medicine, simulation, internet of things, sensors, NASSS framework, qualitative research, document analysis

## Abstract

A digital twin is a computer-based “virtual” representation of a complex system, updated using data from the “real” twin. Digital twins are established in product manufacturing, aviation, and infrastructure and are attracting significant attention in medicine. In medicine, digital twins hold great promise to improve prevention of cardiovascular diseases and enable personalised health care through a range of Internet of Things (IoT) devices which collect patient data in real-time. However, the promise of such new technology is often met with many technical, scientific, social, and ethical challenges that need to be overcome—if these challenges are not met, the technology is therefore less likely on balance to be adopted by stakeholders. The purpose of this work is to identify the facilitators and barriers to the implementation of digital twins in cardiovascular medicine. Using, the Non-adoption, Abandonment, Scale-up, Spread, and Sustainability (NASSS) framework, we conducted a document analysis of policy reports, industry websites, online magazines, and academic publications on digital twins in cardiovascular medicine, identifying potential facilitators and barriers to adoption. Our results show key facilitating factors for implementation: preventing cardiovascular disease, in silico simulation and experimentation, and personalised care. Key barriers to implementation included: establishing real-time data exchange, perceived specialist skills required, high demand for patient data, and ethical risks related to privacy and surveillance. Furthermore, the lack of empirical research on the attributes of digital twins by different research groups, the characteristics and behaviour of adopters, and the nature and extent of social, regulatory, economic, and political contexts in the planning and development process of these technologies is perceived as a major hindering factor to future implementation.

## 1. Introduction

Digital twins—a ‘virtual’ replica of a complex system which is iteratively updated with real-time data—are increasingly being seen as a solution to the present cardiovascular disease crisis [1], a problem much exacerbated by COVID-19, austerity policies, and increasing health and social inequalities [2]. This enthusiasm is widespread in journals, articles, and media, frequently conveyed through the metaphor of promise and potential that encapsulates the sense that new insights will be generated and better medical decisions will be made, resulting in better outcomes for patients. As one leading science and technology media outlet *ITProPortal* declared: *‘digital twins show great promise in health care. By synthesizing and tracking patient data through medical records, medication history, imaging studies and patient-provided health data, healthcare professionals have the potential to […] synchronise all sources of data to create a complete picture of a patient’s needs’* [3].

As cardiovascular diseases (CVDs) are the leading cause of death worldwide [4], it is no surprise that science and technology research in this area leans heavily towards the development of digital twins of the cardiovascular system. For example, within the industry, French software company Dassault Systèmes is in the process of developing a digital twin of the heart to improve device placement and surgical interventions [5,6,7,8], while Philips are developing a digital twin of the heart to better predict heart disease and better tailor treatment and prevention plans to the unique characteristics of individual patients [9]. In UK academia, the University of Sheffield [10] and King’s College London [11] are also developing heart digital twins, leading key research in the area. It is clear: digital twins have been identified as a promising tool for improving clinical decisions and treatment efficacy and safety in the field of cardiovascular medicine (e.g., [12,13]). However, there is also dispute, contention, and caution among these research communities. A mapping review of digital twins being developed for cardiovascular disease by Coorey et al., [14] pushes us to think behind the veil of promise and narrative of perfectly designed technical systems common in the engineering discourse, and reminds us that there is also a human and social dimension that goes into developing and implementing the system as well as its acceptance:
The promise of an active digital twin of either the human heart or the human organism for clinical decision-making remains futuristic. Its advancement holds feasibility and implementation challenges that are not unique to CVD, namely that although grounded in AI it hinges on many human factors.(Coorey et al. [14])

A dose of reality is that to be implemented, complex medical technologies must be accepted as workable and satisfactory by their potential users and must be flexible enough to be embedded into actual practice in the networks of social relationships that make their use meaningful [15]. Many cardiac models have been developed for cardiovascular disease over the last 50 years, yet only a small number of models have reached clinical implementation [11,16]. These lacked validation, lacked clinical interpretability, lacked transparency over the model’s failure rate, did not provide enough evidence of preliminary findings, and failed to produce efficient testing strategies [16]. Corral-Acero et al. [11] further note that regulatory bodies’ demands for rigid assessment of algorithmic performance and adherence to different measures of quality control is likely to slow down future adoption. More specifically, successfully implemented medical technologies must effectively address key challenges surrounding barriers to participation and adoption by clinical experts who may see themselves as working for the public good—thus raising salient issues for the ongoing building and maintenance of the system, such as having trust in the system (or the organisations and/or individuals with whom it is connected) [15]. The aim of this article is to analyse the potential challenges raised by digital twins in cardiovascular medicine, particularly focusing on facilitators and barriers to adoption. The challenges facing the introduction of new technologies within organisational and professional contexts have been widely discussed in the academic literature in the context of innovation adoption frameworks. While several frameworks focus on the dimensions of technology use in organisational and professional contexts, such as Fixsen et al.’s [17] ‘6 Functional Stages of Implementation’, Holden and Karsh’s [18] ‘Technology Acceptance Model’, and Rogers’s [19] ‘Diffusion of Innovations’, empirical work tends to focus on the end user’s perceived utility and ease of use of the technology, highlighting the relevant technical properties of the system necessary to achieve acceptable adoption rather than the whole complex sociotechnical system that combines technical and institutional infrastructures with people who make critical and highly contextual decisions. 

However, two frameworks emphasize the sociotechnical dimensions of innovation adoption. Kotter’s [20] ‘Eight steps to transforming your organisation’ highlights a variety of social challenges, such as forming a powerful enough guiding coalition (making sure that there are enough senior people around to ‘buy in’ to the concept or technology), communicating a vision (making sure that the organisation communicates clearly its vision in which it needs to move in), and changing organisational culture (making sure new approaches become institutionalised as social norms and shared values). Greenhalgh et al.’s [15] ‘Non-Adoption, Abandonment and, Challenges to the Scale-Up, Spread, and Sustainability of Health and Care Technologies’ (‘NASSS’) framework consists of seven domains, including the target health condition(s), the technology, the adopter system (patients, clinical staff, technical staff, administrative staff), the organisational elements, and the wider system of enablers (regulatory or professional bodies, general public). Whilst it is not the only way to approach the analysis of complexity in technology implementation, NASSS is a health-focused framework that has been designed to identify important insights into how healthcare innovations are adopted and has the potential to generate multi-level accounts within each of its domains. For example, ‘dimension 2: the technology’ provides a rich and multi-facetted picture of the technology, including its ‘material features’, ‘type of data generated’, ‘knowledge needed to use’, and ‘technology supply model’. We will discuss each of the NASSS domains in detail in the next section. All in all, while the NASSS framework has much in common with previous adoption frameworks like Holden and Karsh [18], it moves beyond the use of such frameworks in three ways: first, it allows us to consider adoption as a process that spans the entire life-cycle of the technology, including different roles and sectors (i.e., the sociotechnical system); second, it has been designed with the implementation of health technology and health settings in mind; and third, in a study of technology adoption, the framework highlights how challenges during the development phase may impact the implementation phase. It is therefore a suitable framework to help define the dynamics of successful implementation of digital twins in cardiovascular medicine. 

We, therefore, use the NASSS framework to address the question: What are the facilitators and barriers facing the implementation of digital twins in cardiovascular medicine, and how can the barriers be addressed? This question is motivated by an ongoing debate about why technology adoption efforts fail [20] and the lack of uptake of complex systems in medicine [21], which often mirrors the complexity of the condition or disease itself [22]. We argue that digital twins in cardiovascular medicine present a new set of challenges to adoption. Analysis revealed much promise surrounding the use of digital twins in cardiovascular medicine, yet multiple areas of complexity also appeared in our analysis when we identified potential barriers in the literature that related to this technology. For these systems to stand a chance of being implemented, we must look beyond the technical properties of technologies, and broaden our focus to include the actual social mechanisms by which professionals trust, adopt, and use technologies. We demonstrate the dynamic sociotechnical system into which digital twin technologies are likely to become embedded, highlighting key facilitators (e.g., a demand for preventing cardiovascular diseases) and key barriers (e.g., being able to collect and integrate in real-time the multiple information of an actual person into a twin) to implementation in this approach. This study is the first to use the NASSS framework to identify the barriers and facilitators of the implementation of digital twins in cardiovascular medicine. Our application is prospective and anticipatory, responding ahead of time, accounting for the material, social, technical, and organisational challenges in delivering the technology in clinical settings. As medical digital twins come to occupy a significant position in determining national research priorities and the ‘promise’ of digital twins becomes more prevalent in mainstream medical discourses as a common resource for personalised medicine, little or no research exists to support or critique this position. Therefore, it is increasingly important we turn more attention to these key discourses and documents at early stages of a technology’s development process in order to better prepare as well as to guide researchers when ‘implementing’ their technology. The novel contribution of this research is to identify primary barriers and facilitators of medical digital twin technology during the very early stages of its development using the NASSS framework to help future researchers, innovators, and organisations better prepare for the challenges implementation presents and guard against undesirable futures.

## 2. Methods 

The NASSS technology implementation framework was used as an analytical framework [15]. This framework was chosen because it is designed to help researchers categorise the facilitators and barriers of technology implementation. This allows a rich and multi-level account of the different components needed for implementation [23]. This article employs the prospective approach of the NASSS framework, rather than retrospective. That is to say, the approach was to predict and anticipate the possible facilitators and barriers to implementing a new medical technology into routine clinical practice. Such an approach chimes with the principles of Responsible Innovation (RI) by offering forward looking approaches for achieving ‘beneficial’ societal outcomes or on how to stimulate the ‘right’ processes to achieve these goals [24]. The NASSS framework comprises 7 domains (Figure 1), each representing a highly complex ecosystem. According to Greenhalgh et al. [15], it is important not to regard these domains individually, but as being part of a wider sociotechnical system with multiple interactions with each other over time. Each domain includes several sub-domains which are said to surface different kinds of complexity in the proposed programme, highlighting how technologies have a wide range of influences which generate particular activities in particular contexts [22,23,25]. Domain 1 is the condition (for example, an illness such as cardiovascular disease). Complexity occurs when the condition is associated with comorbidities (most commonly in older people aged over 65) and strongly influenced by socio-cultural factors (e.g., poverty or race and ethnicity) [4,26].

Domain 2 is the technology. Complexity may relate to its Material Features (e.g., technical functionality, reliability, speed) [27], its technology features (all digital twins foreground the technology feature of producing in silico models to help clinical experts simulate heart health [6,11] and experiment ‘what if…? scenarios [7,28], the knowledge needed to use it (e.g., all digital twins require a new specialised skill set) [7,29,30], the type of data generated and its management as well as the privacy risks that are brought into play) [12,31,32,33]. Domain 3 is the Value Proposition—both supply side (value to the developer) and demand-side (value to the patient, healthcare system, and taxpayer or insurer). Complexity in this domain relates to challenges in crafting a clear business case for developers, cost effectiveness, and that there is clear desirability for the technology [6,34,35,36].

Domain 4 is the Adopter System: the staff who will be using and maintaining the digital twin as well as the patients and staff who will be expected to use sensors (but who may refuse to use it or find they are unable to use it). Complexity, may arise, for example when there is ‘resistance’ to using new technologies among healthcare staff or patients [23,37,38]. 

Domain 5 is the healthcare organisation or organisations. Complexity in this domain may relate to the organisation’s general capacity to support innovation (for example, strong leadership, sound decision making, and effective human resource management, especially in terms of staff motivation, regular retraining and support [23], and extent of work needed to implement changes (including software maintenance or adaptation) [14,39,40]. 

Domain 6 is the wider system, including the policy context, support from regulatory or professional bodies, and public perceptions. This domain highlights how external factors can influence the uptake (and deceleration) of innovations, such as social, political, technological, and economic factors [13,14,41,42].

Domain 7 is the continuous embedding and adaptation over time (of both the technology and the service or organisation). Complexity in this domain may arise from the technology’s lack of potential to adapt to changing context or from the organisation’s lack of resilience through learning and adaptation [22].

## 3. Results

We applied the NASSS framework to the literature on digital twins in cardiovascular medicine and digital twins in medicine more generally. Our focus is on the domains outlined by Greenhalgh et al. [15], one of the most systematic frameworks for analysing the sociotechnical dimensions of innovation adoption in the medical sector. 

Specifically, the literature that we drew from was a mixture of peer-reviewed publications identified from Medline, Scopus, and Google Scholar databases. We included a variety of articles published in English. These included: research articles, research reports, commentaries, book chapters, and media articles. No limitations were set for the publication date. These databases were searched using a combination of relevant keywords including: ‘medical digital twins’, ‘healthcare digital twins’, ‘cardiovascular digital twins’, ‘medical digital twins and sensors’, and ‘healthcare digital twins and sensors’. Publications referring to any type of medical digital twin (i.e., digital twins that enable personalised medicine, and digital twins that enable strategic planning) were included. Due to the absence of medical digital twins being used for personalised medicine in the real-world, we decided to include medical digital twins for strategic planning in this search.

The first author (PW) conducted the literature search by examining the titles and full-text publications concurrently, and key information from publications which met the eligibility criteria were extracted (e.g., ‘medical digital twins’, ‘cardiovascular digital twins’, ‘medical digital twins and sensors’) and summarised in a table. Next, the NASSS domains served as an analytical framework, and by using deductive principles the content in the table was categorised as a barrier or facilitator and sorted into one of the seven NASSS domains. The analysis was led by the first author, with the second author (TC) supporting the analytical process and providing verification of categories and themes. The organisation of the categories and themes in the table were checked and modified throughout the analysis via an iterative process. The table was then broken up into the seven domains to help bring clarity to our analysis.

In a final set of the analysis, we analysed the complexity of each domain in three levels: *simple* (few components, predictable), *complicated* (many components but still largely predictable), or *complex* (many components interacting in a dynamic and unpredictable way) [15]. In this assessment, perspectives of the authors and findings from outside the specific area of medical digital twins were included in our discussion section to enrich our discussion. We found the existing literature on sensor technologies for health management and monitoring to be especially useful when writing the discussion section given the lack of research on sensors in the context of medical digital twins (there are some exceptions, see [43,44]).

### 3.1. The Condition

#### A Demand for Preventing Cardiovascular Diseases

Table 1 shows the first of the seven domains: the condition. An analysis of the literature on digital twins in cardiovascular medicine reveals how cardiovascular disease is the number one cause of death globally, costing an estimated 17.9 million lives each year—almost one third of all deaths worldwide [4,11,14,26,45]. Cardiovascular disease includes all heart and circulatory diseases, including coronary heart disease, angina, heart attack, congenital heart disease, hypertension, stroke, heart failure, peripheral arterial disease, and vascular dementia [26]. In the UK, demographic shifts towards an older population (who suffers from more cardiovascular issues) and the continuing burden of disease among the population has served as a foundation for the National Healthcare System *Long-Term Plan* report, making heart and circulatory disease a priority [45]. The report makes it clear that heart and circulatory diseases cause nearly a quarter of all early deaths in the UK, yet progress in tackling early deaths has stalled [46].

With increases in cardiovascular diseases, clinical experts are looking to digital twins to help enhance their diagnostic and monitoring toolkit in order to help prevent and reduce the risk of people suffering cardiovascular diseases [1,11,14,39,47,48,49]. This rise in cardiovascular disease is facilitating the development of digital twins in cardiovascular medicine. Second, an analysis of the literature revealed cardiovascular disease to be a complex condition with multiple physiological, socio-economic, and socio-demographic risk factors contributing to the disease (e.g., genetics, dementia, poverty, cultural beliefs, drug dependency), making accurate and personalised risk assessment with tailored treatments the prerogative of doctors in cardiovascular medicine. It is primarily this challenge—the complexity of the disease—forming a highly heterogeneous condition of four main types, such as coronary heart disease, stroke, peripheral arterial disease, and aortic disease (each one having its own symptoms and treatments), that has facilitated the development of highly specific twins of the heart [14]. Indeed, the literature has revealed what types of cardiovascular diseases are of pressing concern to cardiologists, revealing a variety of specific cardiovascular disease-related clinical problems. In much of the literature on digital twins and cardiovascular disease, digital twins for detecting and preventing Ischaemic Heart Disease (IHD, also known as coronary heart disease) are proposed as one of the core models for what is one of the top ten causes of death in the Western world [1,11,14,49]. For example, Coorey et al. [14] indicate a concentrated effort by cardiologists to foster the potential of digital twins to tackle IHD, highlighting how it surpasses all other types of cardiovascular disease “as a cause of premature mortality, with access to, and adoption of, proven treatments being context specific”. Similarly, Corral-Acero et al. [11] also identify ischaemia in patients as a complex clinical problem to solve, pointing out how its underlying mechanisms are poorly understood and a ‘grey zone’ continues to exist in clinical decision making for effective identification and diagnosis. Martinez-Velazquez et al. [49] further present their ‘cardio twin’ model for IHD detection, designed to ‘run on the edge’ to help in the event of a Myocardial Infarction (MI) (a type of IHD) in which arterial blood to the heart muscle is interrupted. To a lesser extent, other cardiovascular disease-specific twins include aortic valve disease [50], congenital heart disease [8], heart failure [51], and abdominal aortic aneurysm [52].

Alongside these efforts, other research looks to better understand the underlying mechanisms of more general human heart function and the origin of cardiac disease as a platform to guide the design of other applications [5,47,53,54,55]. Baillargeon et al. [5] working on behalf of the Living Heart Project, are in the process of developing a heart digital twin to simulate the blood flow between the four different chambers of the whole heart, attempting to visualize the mechanical response of the living human heart throughout its cardiac cycle. Underscoring the need to address the rise in cardiovascular disease, studies in the UK, the USA and other jurisdictions are looking to digital twins to help tackle the disease as it continues to remain the leading cause of death in developed countries and exhibits a high socioeconomic cost [8]. Given the complexity and heterogeneity of the disease and the fact that its origin, progression, and treatment move beyond physiological characteristics to include context-specific risk factors, such as demographic make-up, ethnic heterogeneity, and economic status [56], digital twins are perceived as the only way to truly know someone’s heart health.

### 3.2. The Technology

#### 3.2.1. Technical Functionality and Reliability 

Table 2 shows the second of the seven domains: the technology. The functionality of the ‘digital’ twin is based on a communicative situation that is reliant on a synchronised cyber-physical system between the primary real-world context (i.e., the ‘real’ twin) and a secondary virtual context (i.e., the ‘digital’ twin) [14,27,57,58,59,60,61,62,63]. A synchronised cyber-physical system is seen as an important facilitator because it allows the digital twin to mature through timely and accurate correspondence between its real entity. Constant synchronisation between the twins may give clinical experts confidence over the quality of their model because it allows the virtual twin to quickly update as it adopts the properties of its physical twin, enabling what is known as ‘closed-loop optimisation’ [14]. 

Secondly, it was also seen as an advantage that the digital twin platform required connection to the Internet of Things (IoT) [12,14,28,33,43,59,60]. The statement below from Barricelli et al. [59] indicates that IoT based solutions are likely to become key to digital twin implementation because they facilitate the interrelation between different components, such as sensors and actuators for the transfer and collection of data:
The advent of the Internet of Things (IoT) is changing the way data are exchanged among different sources. Indeed, the diffusion of technologies, such as (embedded) sensors and actuators connected through the Internet, allows a continuous exchange of Big Data.(Barricelli et al. [59])

This approach, otherwise known as data-driven smart manufacturing (BDD-SM) uses the IoT and sensors to produce and transport Big Data [59]. In this instance, the IoT is considered as having multiple purposes. For example, the IoT is said to encourage the use of online platforms for storing data and increasing access of the digital twin and its data [28]. One of the most common IoT platforms mentioned across the literature is cloud computing and has become seen as the key facilitator of IoT-digital twin implementation [27,28,33,40,43,59,60]. According to Mell and Grance [64] cloud computing is:
A model for enabling convenient, on demand network access to a shared pool of configurable computer resources (for example, networks, servers, storage, applications, and services) that can be rapidly provisioned and released with minimal management effort or service-provider interaction.(Mell and Grance [64])

**Table 2 sensors-23-06333-t002:** Domain 2: The Technology.

Complexity Level	Theme	Facilitator	Barrier
Complex	Technical functionality and reliability	Real-time synchronisation between real and virtual entities [14,27,57,58,59,60,61,62,63]	Establishing a synchronised cyber-physical system [27,57]
		Internet of Things (IoT) [12,14,28,33,43,59,60]	Vulnerabilities in computational storage and infrastructural resources [14,60]
		Cloud computing [27,28,33,40,43,59,60]	Hacking and Viruses [30,34,40]
Complex	Technology Features	Real-time simulation [6,7,9,11,14,29,47,57,59,65] Real-time remote monitoring of a patient’s health [14,27,28,33,39,40,43,60,66,67,68,69]	Lack of regulations and rules establishing validation methodologies [14,57,59]
		Real-time experimentation [7,11,28,55,57] Predictive maintenance [9,13,14,27,32,70] Virtual collaboration [28,40,67,71,72]	Specialised skills required [7,29,43,44]
Complex	Patient data	Real-time data collection [3,30,31,32,33,43,44,60,70]	Huge amounts of data required [7,32,57,63,66,69,73]
		Data fusion techniques and multiscale data integration [27,28,40,59,74] Dedicated structured reading team [75]	Challenges of integrating data from different sources [27,28,40,42,59,75,76] High cost and time consuming [40] Privacy risks: (1) commodification [31,33,39]; (2) surveillance [33,34,60,76,77]; (3) behavior manipulation [12,31,33,34,39,60]

Cloud computing is then imbued with expectations inseparable from the IoT paradigm. Analysis has yielded significant insight into the potential of cloud computing, arguing that one of the main benefits of cloud computing is that patient data can be collected from sensors and sent to the ‘cloud’ automatically via a wireless network that clinicians can access for processing and analysing [6,27,33,40,43,44,62]. Liu et al. [27], discussing the potential of their Cloud Digital Twin Healthcare (Cloud Digital TwinH) system to support the self-management of elderly people, argue that once data are uploaded to the cloud it “offers new possibilities such as ubiquitous access to medical data, and opportunities for new business models”. Indeed, the promise of ubiquitous access is seen as a key facilitator to digital twin implementation enabling other people or systems outside of the local digital twin healthcare team to access the digital twin regardless of their location, facilitating date exchange, and knowledge sharing [27,40]. Another study sheds further light on the dynamics of cloud-based applications by exploring how cloud algorithms can quantitatively detect frailty and pre-frailty in people over the age of 60 in order to return them to “robustness” [43]. Despite its promise, however, there are currently no cloud healthcare platforms in operation that support the collection and real-time monitoring of patient Big Data because they are still in development [59]. Moreover, cloud platforms enable automatic real-time data collection and reduce the possibility of human transcription errors [33]. Crucial here is the promise of ‘real-time’ data collection from sensor technologies and will be discussed in more depth later in Theme 3: ‘Patient Data’.

However, efforts to establish a synchronised cyber-physical system that is supported by the IoT raises a variety of challenges to adoption. For example, the effort to maintain a virtual communication network between real and virtual entities in real-time highlights four challenges. First, there is the challenge of ensuring that these models are verified and validated before they are used clinically to ensure that they are correct and accurate [14,57,59]. Second, there is the challenge of ensuring that adequate calibration strategies are in place to ensure the digital twin runs synchronously and evolves in parallel with its physical organ [27]. Third, there is the length of time it may take to achieve fully functioning synchronisation. The fourth challenge leads us into cyber security challenges and risks of hacking and viruses [30,40]. According to Miskinis [30], digital twin implementation opens up three main opportunities for hacking, these include: hacking into the sensor device; hacking into the synchronisation between digital twin and device; and hacking into the digital twin system itself. Fuller et al. [40] also consider the vulnerability of cloud platforms to hacking and viruses, arguing how cloud infrastructures require “robust security”, pointing out how more scrutiny, regulation, and measures concerning digital twins are needed in order to ensure personal data are protected. In the analysis of these issues, medical digital twins have opened up new approaches to breaches in health data, revealing how the data stream between digital twin and monitoring technologies can be hacked or interrupted [30]. Here, then, is an added incentive for increased efforts to produce and distribute personalised surveillance techniques in order to ensure data integrity and security. However, this raises a further issue: the possible scope of state surveillance and an example of how digital twins can be used for ever-more intrusive surveillance practices [34]. This latter issue will be discussed in Theme 3: ‘Patient Data’.

#### 3.2.2. Technology Features 

The feature of running a dynamic real-time simulation is perhaps what distinguishes the digital twin from other virtual models in medicine. A real-time simulation that is continuously updated based on changes in the object it represents (e.g., a real human heart) is currently lacking from modern applied approaches of virtual models in medicine. The literature notes that real-time simulation of the heart will use algorithms to predict real-time changes of the real-world heart which correspond to its virtual counterpart [6,7,9,11,14,29,47,57,59,65]. As such, digital twins will be continually trained and updated based on real-time data as well as the historic or retrospective medical data of the patient. When talking about real-time simulation in this context, there is excitement at the prospect of the monitoring of a person’s physiological condition in real-time and remote detection of disease/illness [14,27,28,33,39,40,43,60,66,67,68,69]. Hose et al. [67] take up this unique feature and imagine a future in which medical digital twins provide clinical experts with the ability to produce characteristic and diagnostic measurements to monitor the effects of all types of interventions, “from lifestyle through to medical and surgical options”. It is imagined that the real-time monitoring of patient health could potentially lead to improvements in prognostication, diagnosis, and more targeted interventions and treatment strategies, not only an improved model. The remote monitoring and detection of physiological signals should, at least in theory, bring a patient’s health status closer to clinics to form new ‘personalised’ treatment options. The ability to collect, process and simulate real-time data is seen as an advantage because it allows for model-based personalisation of patient-specific physiological and behavioural changes (including blood flow, mechanics, and electrical impulses) [39].

A second feature of a digital twin application is the ability to carry out in silico ‘experiments’ [7,11,28,55,57]. This feature holds the promise of producing experimental manipulations that go beyond the single mode of simulation. For example, Dassault’s *Living Heart Project* is said to have produced a software that allows different experimental modes to be used by clinical experts, allowing them the opportunity of bringing in additional factors to see how the heart behaves under different conditions. According to Mussomeli et al. [28] this experimental set up allows clinical experts see something different with different tools to understand ‘what if?’ scenarios more clearly. In Scoles’s [7] commentary on the software, for instance, the clinician is described as being engaged in different modes of experimentation:
The user can manipulate it—stick in pacemakers, reverse its chambers, cut any cross section, and run hypotheticals […] medical simulators see both inside and into the future of virtual organs, without needing to put on scrubs. (Scoles [7])

Similarly, another project based in King’s College London involved in the development of a personalised heart are also interested in discovering new knowledge about how a digital twin of a patient’s heart will behave under defined scenarios, especially in terms of its ‘stiffness’:
We already extract numbers from the medical images and signals, but we can also combine them through a model to infer something that we don’t see in the data, like the stiffness of the heart,’ Lamata says. ‘We obviously cannot touch a beating heart to know the stiffness, but we can give these models with the rules and laws of the material properties to infer that important piece of diagnostic and prognostic information. The stiffness of the heart becomes another key biomarker that will tell us how the health of the heart is coping with disease.(Lamata [55])

Lamata [55] states how having a digital twin of the heart may allow clinicians to *‘infer something that we don’t see in the data, like the stiffness of the heart’*. In stating this, he clearly links the model with the capacity for experimentation. It is clear that experimentation is common to all threads: they are made of scenarios that are all hypothetical in nature. For example, in a position paper by Corral-Acero et al. [11] for a digital twin to enable precision cardiology, we find significant attention being paid to scenario development to help predict how a patient will respond to a specific therapy (e.g., adverse drug reactions) and identify sub-populations at risk of cardiovascular disease. Together, having a digital twin of a patient’s organ allows clinical experts to virtually analyse their patient’s health and plan therapies safely in a virtual space before being tested in the real-world [32]. The turn to simulation and experimentation, as we have highlighted above, is a significant driver behind digital twin implementation.

A third feature that may facilitate adoption is the technology’s capacity to receive continuous updates about the functionality and structural integrity of its sensors [9,13,14,27,32,70]. According to Baily [70], the promise of ‘predictive maintenance’—an engineering concept—to reveal the device’s state in real-time is seen as something that would facilitate digital twin implementation:
Proactive remote monitoring allows healthcare leaders the opportunity to rectify impending issues remotely and to schedule maintenance by a service engineer as appropriate, impacting the fewest number of patients.(Bailly [70])

This feature is likely to help clinical experts predict the ‘mechanical health’ of its sensor technologies because it monitors the functionality and structural integrity of sensors. Clinical experts, here, are uniquely positioned to ‘do’ maintenance work or call on the help of more technical experts because they are able to detect if there is something wrong with the structural and mechanical health of the sensors and predict when technical interventions might be useful before issues occur [13]. The consequences for this, as suggested by Baily [70], means maintenance can be better managed in terms of smaller disruptions to patients and ultimately a more flexible workflow. Jeske [33] recognizes how the promise of predictive maintenance has the potential to enhance the lifecycle of the technology and therefore potential cost savings.

A fourth feature that may impact adoption is the opportunity for virtual collaboration [28,40,67,71,72]. Compared to in-person collaboration, clinical teams will be able to collaborate with experts outside the traditional bounds of clinical departments—e.g., biology, chemistry, physics, engineering, and mathematics—creating trans-disciplinary approaches that allow the collaboration of data and scenario-based decision making on a broader scale. Therefore, digital twins have the potential to provide clinical experts and digital twin developers the flexibility to include different types of knowledge and styles of reasoning, enabling “virtual collaboration” that can best serve patients [28]. Going one step further in such virtual collaboration is the promise of combining digital twins with other online technologies, potentially building a global ‘Internet of Medicine’ which could become the “conduit for a global healthcare ecosystem” [72]. Virtual collaboration further broadens that picture by creating new visual and digital standards and new commercial opportunities in visual and interactive tools, algorithms, and concepts [28]. 

However, despite these facilitators, the ability to analyse simulations and to ‘do’ experiments is likely to form expertise barriers to the technology’s implementation [7,29]. Scoles [7], in particular, warns how without technoscientific training clinical experts are unlikely to adopt and “buy into” the technology. This also raises questions about the skills and training required of clinical experts in terms of understanding the technical functioning of sensors used by patients. Similarly, patients also require training in using their sensors and monitoring technologies [43,44], which we will return to later in this article.

#### 3.2.3. Patient Data

Digital twin users are aware that they need to capture as much patient data as possible and that this information must be acquired from a variety of different sources, including real-time sensor technologies and retrospective patient records. Digital twins are said to have the potential to collect patient data in real-time through IoT devices with digital sensors, such as smartphones and wearables (such as ‘Fitbit’ or the ‘Smartwatch’) [3,30,31,32,33,43,44,60,70]. These sensors can collect a variety of measurements ranging from walking speed, to heart rate, to body temperature and are seen as a more efficient method of data collection to the collection of post hoc data [63], such as blood tests, or scans which are “expensive time-consuming tasks” [30]. Similarly, it is believed that the abundance of large clinical datasets such as Magnetic Resonance Images, Electronic Health Records (EHRs) and Hospital Episodes Statistics (HES) will be included in the model, enabling more accurate and efficient models that reflect the state of the patient [55,73]. 

A second facilitator to the implementation of digital twins comes in the development of new data fusion techniques and multiscale data integration. For example, researchers have been drawn to developing new methods for effective integration of diffuse data from different sources in order to produce more consistent, accurate, and useful information [27,28,40,59,74]. This process of integration calls on strategies to be put in place for data cleaning, which may use ‘data fusion algorithms’ (e.g., neural network, Kalman filter, etc.) and ‘high-dimensional data-(de)coding and analysis techniques’ to ensure data integrity and consistency [27,59]. However, standardised approaches to data fusion are currently lacking in research and innovation of digital twins [40,42]. One of the main points at issue here is the way patient data needs to be integrated into local human and organisational ecosystems and workflows from decentralised data sources [28,42]. The second challenge in the integration of data is making sure that it is cleaned in a way that digital twins are made of high-quality data [40,75,76]. Fuller et al. [40] say that it is important to ensure that data are not of “inferior quality” and can be sorted and cleaned to ensure only the “highest quality of data is fed into the AI algorithms”. The third challenge of data fusion techniques and multiscale data integration is a sensitivity to whom or what gets collected. 

Despite advances in the increasing use of statistical methods for patient anonymisation to protect sensitive information or the identity of the data owner [74], challenges of anonymizing data in the process of integration remain. Much of the literature pointed out privacy risks as a barrier for implementation in the health domain [12,31,33,34,39,60,77]. Jeske [33], in particular, has identified potential privacy risks that stakeholders need to give special attention to, pinpointing commodification of personal (health) data, surveillance, and behaviour manipulation as three main risks which have the potential to invade the individual’s privacy. First, Jeske reveals the risk of commodification and how healthcare providers have become valued for the increasing commercial or research value they have for external organisations (such as businesses, the government). *The Medical Futurist* also make this point with the following thought, leading what they call a “stall in the progress of digital wins in healthcare”:
This will become an issue when such companies make millions of profit out of patients’ data. The patients themselves likely won’t receive any cut and will have to pay to have access to this service. This could lead to a major backlash from patients and other stakeholders. In the worst-case scenario, it will lead to a stall in the progress of digital twins in healthcare. (Medical Futurist [39])

Likewise, when Copley [31] addresses the promise of medical technology companies developing digital twins, she reveals privacy concerns of patient data being exploited and sold by for-profit companies without compensating the patient. She points out how patient data could be used primarily by companies as a commercial tool and implications over the ownership of data is a *‘gray zone that could lead to a patient backlash if companies start making fortunes from it’*. The second privacy risk related to healthcare digital twins is surveillance, raising the risk of state surveillance and how digital twins can be used for intrusive surveillance practices [33,34,60,77]. Surveillance here is related to the presence of personal health from a patient’s everyday life and to potential mass measuring of health data in homes. According to Maeyer and Markopoulos [34], surveillance could result in the “social sorting of a population … resulting in inequality in certain populations”. The third privacy risk related to digital twins, is the potential for “behaviour manipulation”, whereby personal data collected by a system or platform is able to generate behavioural predictions about that individual [33]. These ‘behaviours’ can then be sold to third parties (such as advertising companies) for the purposes of influencing or managing those whose data has been gathered [33]. Once again, this is not to suggest that digital twins or clinicians will unwittingly give access to untrained individuals or external organisations, but simply that we must discuss who should and must have access to digital twins [60,76,77].

Despite the increasing availability of patient data and increasing sophistication of tools for anonymisation, there may still not be enough data to ensure the optimal running of a digital twin. The relationship between AI-driven models and data is not without friction in the development and implementation of digital twins, with some commentators highlighting that there is simply not enough data around in order to develop or implement digital twins successfully [66]. For example, companies such as Siemens explicitly state on their website how a substantial amount of data are necessary to develop an effective digital twin of the heart: *‘Siemens Healthineers is developing algorithms that generate digital models of organs based on vast amounts of data’*: [73]. The following excerpt taken from *Slate Magazine* journalist Sarah Scoles [7] on the topic of digital twins in cardiovascular medicine exhibits this demand for data in the context of its founder Steve Levine’s story about his daughter’s rare heart condition:
Inside his daughter’s heart, the left chamber is where the right one should be, and the right one is where the left should be. Because of that reversal, Jesse received her first pacemaker at age 2. But the wire leads kept breaking, and by age 20, she was on her fourth device. Her doctors said that to understand her rare condition and come up with better treatments, they needed more data. And that data would be difficult to retrieve. Given all of the metal in her chest, they couldn’t give her an MRI. So Levine decided to create data about his daughter—digitally. And so in 2013 he and Dassault—which had recently branched into life sciences—decided to apply Simulia’s technology to the heart.(Scoles [7])

To understand Levine’s daughter’s heart condition and, specifically, how her pacemaker leads kept breaking, the clinicians highlighted how ‘*they needed more data.*’ In recognition of what is becoming an important trend in modern medicine, Levine’s ambition was to build software in order to *collect* new data. Although Levine is referring to Dassault’s SIMULIA software, his vision would equally apply to the various digital twin projects around the world using huge amounts of patient data to develop computational models of a particular real-world artifact [32]. As this quote suggests, the vast collection of patient data is framed as a route for building the SIMULIA software that concerns the human heart. Scoles [7] reports that processing patient data offers the promise of unlocking new insights and accelerating breakthroughs in treatment, but this is coupled with a shortcoming about how this data *‘would be difficult to retrieve’.* Scoles touches on a much wider debate within AI development about how patient data are collected, standardised, and curated. Scoles suggests then, that while there is significant data available for clinicians—a critical element in the training of digital twin algorithms—the collection of new data that is medically relevant to the patient may prove challenging for those involved in developing and implementing digital twins. 

A third factor facilitating the implementation of digital twins in cardiovascular medicine may come in the form of a “dedicated structured reading team” [75]. Siemens Healthineers have recruited a data science reading team in order to build a database that can facilitate access to, store, and process massive amounts of data from different sources:
Over the course of the last few years, Siemens Healthineers has invested in a dedicated structured reading team, building a database that can potentially access more than 750 million curated images, reports, and clinical and operational data which are fed into and also used to train algorithms.(Siemens Healthcare GmbH [75])

In such a case, a designated reading team will cover the process of data collection and are key players behind the integration of data from different sources related to the digital twin. However, document analysis has highlighted how Siemens Healthineers are an exception because they have been designing and implementing digital twins for different domains (such as aerospace, automotive, and manufacturing) for some time [39] and have substantial resources to devote to the data collection process. Having such a dedicated reading team may not be feasible for many healthcare systems since it would cost a lot of money and time. Further research is needed to explore the role of designated data science teams and experts with data science skills in the digital twin ecosystem. 

### 3.3. The Value Proposition

#### Desirability 

Table 3 is the third of the seven domains: the value proposition. The NHS healthcare system is confronted with a constant rise in demand to diagnose and treat people with symptoms of cardiovascular diseases which is estimated to cost the NHS and wider society in England £15.8 billion a year [26]. This means that there is an immediate requirement to maximise the efficiency of healthcare services but at the same time minimise costs. This desire to prompt faster, more efficient healthcare services in a cost-effective manner was explicitly addressed across industry websites, journal articles and media as significant motivation behind digital twin development and implementation [6,26,34,35,36,58,78]. For example, an article by D’Souza in the American simulation magazine *Benchmark* [6] entitled ‘Technology to Transform Lives’ highlights the impact of cardiovascular diseases and its spiralling costs rising from treatment:
The decision to model the human heart was motivated by several reasons. Cardiovascular disease is the leading cause of death in the US and other developed countries and imposes a high socioeconomic cost. Spending on circulatory conditions [primarily heart disease and hypertension] constitutes the largest portion of US healthcare expenditure. (D’Souza [6])

The D’Souza quote above explicitly states that the prime motivation for the implementation of its digital twin was because of the *‘high socioeconomic cost’* of cardiovascular disease, a narrative that highlights the strong pressure for cost containment within the US healthcare system. Similarly, Rahman et al. [58], in a report titled *Transforming Healthcare with Digital Twins* on behalf of the Challenge Advisory Group, say that:
With healthcare costs rising globally and the world’s population increasing, now is the time to use our digital counterparts to make changes to the system to enable a more efficient solution for both healthcare professionals and patients without causing either any harm. (Rahman et al. [58])

In their report Rahman et al. [58] underline the relationship between cost savings and efficiency, and the potential for saving lives. The report suggests that the movement towards digital twins of hospital departments such as an operating room or accident and emergency is essential to the department’s interest in efficiency, which are also concerns for the benefit of the patient. The result can be improved cost savings and healthier patients:
Implementation of digital twins in hospitals, offers numerous benefits. It has been reported that digital twins could provide a 900 percent cost in savings in hospital and a 61 percent reduction in blue code hospital events. (Rahman et al. [58])

A digital twin can act as a cost saving tool or efficiency saving technology in the healthcare environment, with hospital digital twins referenced as providing a *‘900 percent cost in savings’ and a ‘61 percent reduction in blue code’* (i.e., medical emergency) events happening in hospitals. According to Rahman et al. [58] the potential cost saving advantages of this type of digital twin has become evident in Mater Private Hospital Dublin. In a collaboration with Siemens, Mater Private Hospital aimed to tackle increasing demand for medical images and its rising personal costs of producing medical images by implementing a digital twin of its radiology department. With the adoption of the technology to simulate actual MRI and CT workflow, the digital twin optimised the turnaround time for producing images which improved patient waiting times (*‘the identified improvement potential was a nearly half-hour reduction in patient waiting time, and significantly reduced staff overtime costs’*: [78]). Many publications shared this perspective, viewing personalised digital twins in positive value terms as supporting better (more personalised) care and making it easier and quicker for clinical experts to diagnose and treat their patients [1,6,11,31,34,35,36,42,58,60].

Taken together, the improvement in patient outcomes while lowering costs in hospitals and departments can be said to facilitate implementation by bringing efficiency gains to existing practices, transforming inefficient practices and delivering socio-economic benefits for all. However, despite there being a small but growing body of research that relates to cost savings and the delivery of high-quality care, evidence is clearly weighted towards digital twins for ‘strategic planning’, such as the development of a digital twin of a hospital for supporting hospital organisation and management [63,78]. This confirms that research which evaluates the cost-effectiveness of digital twins for personalised medicine (i.e., a patient’s heart) to inform clinical decision making and treatment is currently absent. 

There is no empirical information on whether digital twins for personalised medicine are cost-effective. This opens up questions for analysts about costs and expenditures relating to a personalised twin of a patient’s organ or process they represent [35], indicating how the sheer complexity of this type of engineering is likely to incur more significant costs to developers and healthcare systems compared to digital twins of hospital buildings or departments. It also leads to questions of very fundamental features of cost containment, such as how much does an organ-based medical digital twin cost to run while it is implemented, and how much are maintenance costs, where costs vs. benefits need to be weighed into the development and implementation process. 

### 3.4. The Intended Adopters

#### Collaboration 

Table 4 is the fourth of the seven domains: the intended adopters. NASSS analysis exhibits positive attitudes towards the adoption of digital twins and sensor technologies (e.g., they anticipate the value of the data generated, and anticipate the digital twin’s ability to help prevent cardiovascular diseases). At present, these positive attitudes come from intended adopters of digital twins such as cardiologists who are all engaged in a highly collaborative affair with leading digital twin developers, medical device developers, regulatory agencies, and other clinical experts [6,9,69]. However, it is not clear who the other adopters of digital twins and sensor technologies will be in the field of cardiovascular medicine and what, exactly, will be the role of each of these when it comes to implementation? If this is the case, then more research is needed to clarify the roles of all the different staff who can contribute and enable a successful implementation.

NASSS analysis also highlighted clinician–patient collaboration as an important facilitator for digital twin implementation. Digital twin developers and clinicians perceive high ability in patients to use their sensor technologies, highlighting close clinician–patient collaboration as an important facilitating factor for successful adoption [9,33]. According to Jeske [33], closer collaboration with patients may increase the chances of widespread adoption and scale-up of the digital twin. It is in this sense that clinician–patient collaboration is central to the implementation of digital twins, especially when it comes to the system’s sustainable upkeep and the proposition that implementation must be understood as part of a broader sociotechnical project of healthcare intended to include patients [33]. A good example of this point is given by Philips executive Henk van Houten [9], who suggests the inclusion of sensor technologies brings in its wake an emphasis on clinician–patient collaboration and patients as the agents of data collection:
A cardiologist could delegate certain routine measurements to a GP. And, at the discretion of the GP, a patient may be able to perform certain measurements at home, such as taking blood pressure. The digital patient paradigm requires us to rethink the collection of health data as a continuous, collaborative process with the patient at the center. (Van Houten [9])

Van Houten [9] sees this as part of a paradigm shift, now gathering momentum, towards a more collaboratively informed healthcare that puts patients at the centre. In support of Van Houten, research in recent years has provided compelling evidence in support of patient centrism as a “paradigm shift” in healthcare delivery, because of the way it is has restructured relationship dynamics between patient and provider and allowing patients to play a more vigorous role in managing their health [79]. It is now becoming clear that this orientation towards patient-centric principles is linked to greater patient compliance, better recovery and health outcomes, and declined readmission rates [79]. Similarly, the Academy of Medical Sciences [80] recognise that data-driven technologies may cause a “paradigm shift” in the way that healthcare is delivered, which is likely to have implications for the healthcare workforce [81]. For example, who in the workforce will be accountable and responsible for evaluating sensor technologies? What training is required? How will capacity and time for evaluation of these technologies be built into the existing workflows? [80]. It also opens up questions around very important features of help-seeking, such as how to alert or communicate to the patient that there is a health issue, and how much information is communicated to them, including the route(s) to care? This issue is highlighted in the quote below from Bruce Rule [68] from *Karma Magazine*: *‘a patient could wear trackers or sensors that send data to the virtual twin, which would be monitored for any irregularities or stresses. If any occur, the patient would be told to seek care’*. Rule’s [68] quote highlights a key challenge clinicians face when considering how patients seek help. This issue of being *‘told to seek care’* resonates with those identified in previous studies on self-monitoring technologies (e.g., [81,82]) and their liability for any consequences such as adverse health outcomes. For example, safeguards should be foreseen to prevent any unintended adverse impacts such as: what if a patient feels breathless or ill, but decides not to seek help because the device has not flagged an issue? We will return to these later on in the discussion.

We have already mentioned that clinician–patient collaboration involves patient adoption of sensor technologies, requiring patients to use sensors and tools to help build their digital twin [9,33,34,36,43,49,55,68,69,83,84]. For the patients, this can mean clinical experts having to convince their patients about the importance of the sensors, which they justify as an effort to promote a vision of seamless data collection and personalised medicine. For example, Bloomer Tech is developing a bra with sensors that monitor a person’s heart function, while Apple and Johnson & Johnson are developing digital health tools that can detect irregular heartbeats on a person’s Apple Watch [68]. Elsewhere, a team of University of Sheffield researchers looking to build a digital twin of a patient’s heart to help diagnose and treat cardiovascular diseases are looking to collect data from a broad suite of “miniaturised sensors” so that patients can wear them throughout their daily lives rather than just in a clinic or a doctor’s surgery [69]. Although sensors are anticipated to be a key feature in digital twin implementation, no research has explored how to successfully implement these devices. This means there is an absence of empirical research on professional and patient accounts of digital twin use, sensor use and self-monitoring practices in the context of medical digital twins. For example, previous research on the intended adopters domain has highlighted how users can lack the capability or willingness to learn the technology, because it represents a threat to their professional identity or scope of practice [85], or a lack of training, staff continuity, and support may influence a lack of uptake [25]. Although sensor development and implementation, in this context, is still nascent, it is worth turning our attention to existing work on current sensors and self-monitoring technologies that have been implemented in the field of personalised healthcare in order to help us get a sense of potential facilitators and barriers. This will be discussed later in the discussion section. For the time being, however, our NASSS analysis revealed two facilitators and two barriers for the use of patient sensors in the context of medical digital twins.

**Table 4 sensors-23-06333-t004:** Domain 4. The intended adopters.

Complexity Level	Theme	Facilitator	Barrier
Complex	Collaboration	Developer–clinician collaboration [6,9,69]	
		Clinician–patient collaboration [9,33]	Socio-ethical issues concerning privacy and individuality [86]
		Advantages in patient sensors and patient self-monitoring (e.g., patient empowerment) [9,33,34,36,43,49,55,68,69,83,84] Ease of use [34] Training program [27,36,44]	Sensor damage and failure [12,27,28,33,59,77]

Leading the Personalised In silico Cardiology consortium at King’s College London, biomedical engineer Pablo Lamata [55] shows how the use of sensors can empower the self-care abilities of patients:
It is also the vision of people being more empowered and being more in control and aware of the impact of their lifestyle choices in the health of their hearts. We will have more wearables that can monitor aspects of our health rhythm, heart sounds or level of physical activity. This unit is also talking to the digital twin that lives in the hospital.(Lamata [55])

Lamata’s account relates to the patient’s empowerment and how having a digital twin can provide the patient with more control over their daily heart health and therefore lead to better health management and well-being promotion [36].

A second facilitator that is key to patient adoption of sensors is the fact that some sensors can be equipped with a specialised user interface, which make them easier to use [34]. For example, Natural User Interfaces (NUIs) often combine voice, gesture, motion or touch because these ways of interacting are already familiar to humans and these ways of interacting with the technology come very natural to humans [34].

A third facilitating factor for adoption is likely to be the availability of training programs or person-centered digital coaching experience for patients, leading patients to effectively self-manage [27,36,44]. Diaz et al. [44] recognise the need for an “always ready smart coach” on a patient’s smartphone to facilitate the appropriate implementation of a patient’s digital twin. If this is the case, a smart coaching system must provide training to patients who uses sensors in order to perform accurate measurements of the body, such as making adjustments in their posture and performance. Likewise, the clinical experts and staff that oversee the patient’s digital twin and their sensors also need knowledge and training in these tools, with a lack of infrastructure, clarity in regulation and training is seen as another barrier to deployment [27]. 

NASSS analysis also revealed socio-ethical risks concerning privacy and individuality [86] as having the potential to impede the adoption of medical digital twins from the perspective of potential adopters and end users. A substantial proportion of interviewees from the industry, research, policy, and society (including current patients) revealed several important factors such as privacy and property of data, disruption of existing societal structures, inequality, and injustice which may all contribute to intended users (staff and patients/clients and their carers) not using or accepting the technology. While research by Popa et al. [86] has started to fill the void of empirical research about the social and ethical consequences of medical digital twins, it is firmly rooted in peoples’ predictions about the future rather than on the present or the past.

Analysis also revealed a second risk of sensor damage and failure as a barrier for implementation, pointing out how on-body sensors may become subject to defects, damages, and failures [12,27,28,33,59,77]. For example, sensors carried by the patient may be dropped or accidentally stolen and destroyed. In addition, sensors may become subject to “fatigue-damage” with cracks appearing in the structure of the device [59]. Such concerns raise problems of ‘data misuse’ and ‘data loss’ [77]. Despite this, NASSS analysis shows that digital twin developers are considering these issues by equipping their systems with predictive maintenance processes which have the potential to protect their sensors from defects, damages, and failures through real-time monitoring, structural life prediction, and device management. Whilst predictive maintenance processes have been used in different fields such as manufacturing and aviation (for example, detecting and predicting damaged aircraft structures) [59], such a process has yet to be completed at any scale for sensors on patients. According to Jeske [33], sensors fixed straight onto the skin (as opposed to wearables like bracelets or smartwatches) such as an “electronic tattoo” made of graphene allow for “high-fidelity biometric sensing” have less risk of being damaged because they adapt better to the skin’s reaction during movement.

### 3.5. The Organisation

#### Changes to Existing System

Table 5 is the fifth of the seven domains: the organisation. There is some evidence that digital twin implementation will be a long and costly enterprise [14,39,40,57]. Fuller et al. [40] has highlighted how one of the biggest challenges to digital twin implementation is the setting up and maintenance of its high-performance IT infrastructure, in the form of up-to-date hardware and software. For example, Fuller et al. [40] draw our attention to the high cost of installing and running digital twin systems and highlight how implementation is potentially threatened by the costs of high-performance graphics processing units (GPUs) that can range from $1000 to $10,000. The recruitment of a data science reading team that is needed to build a database that can facilitate access to, store, and process massive amounts of data from different sources into the digital twin is also documented as an additional cost. In such a case, a designated data science reading team is needed to cover the typical instances of data collection and are key players behind the integration of data from different sources related to the digital twin [75]. However, having this level of technical infrastructure would come at a high cost. This suggests only the most affluent organisations will be able to install and implement digital twins (and their sensors) or even which patient will be able to afford the technology [39]. Fuller et al. [40] is one of the only studies to consider the cost of implementing a medical digital twin system and what implications a digital twin infrastructure has on the organisation. Further research is needed in this domain.

### 3.6. The Wider System

#### Societal Development

Table 6 is the sixth of the seven domains: the wider system. NASSS analysis did not reveal any support from regulatory and professional bodies on the use of digital twins in medicine given the lack of available evidence that they can be of use in clinical medicine [14,41,57]. Despite the promise of medical digital twins, no regulatory or legal standards have been set out. This presents regulatory and professional bodies with a complex individual and organisational challenge, particularly when legal and matters around AI continues to have its challenges [40]. However, a growing number of studies have started to address the regulatory, legal, and ethical challenges of digital twins for personalised medicine with the aim of anticipating governance challenges [1,6,8,11,14,32,33,40,41,42,64,86,87]. For example, D’Souza [6], working to develop Dassault Systèmes’s Living Heart Model, claims that regulatory authorities can potentially act as a “barrier” to digital twin innovation because of their “intense scrutiny” and “stringent demands” on medical device manufacturers. As an example, D’Souza [6] highlights how the existing stringent regulatory approval process has led to a “culture of caution” causing innovation in cardiovascular medicine to effectively slow down. To overcome this shortcoming, D’Souza [6] proposes the adoption of validated simulation tools and models which stakeholders can use to help establish common processes and best practices. This chimes with Corral-Acero et al. [11] who go one step further. They observe that digital twin implementation for cardiovascular disease requires a coordinated drive from scientific, clinical, industrial, and regulatory stakeholders in order to produce guidelines, gold-standards, and benchmark tests to establish the level of rigour needed for computational modelling (as they allow regulators and other stakeholders to judge computational evidence. 

Given the lack of governance frameworks for the development, deployment and use of digital twins in medicine, the use of simulations equipped with AI has sparked concerns around human enhancement and the lack of ethical mechanisms for safeguarding the human rights of patients that have digital twins [32]. This complex matter around human enhancement as a potential significant barrier to regulatory approval and implementation will be discussed next. According to Bruynseels et al. [32] research on the way in which medical digital twins are being developed incorporate the traditional bioengineering model which sees technology as applied science (see, for example [11,47]). Bruynseels et al. [32] point out how researchers in biomedical engineering have an essentialist view of human health; where cultural ways of knowing are to be explained and subsumed in deterministic bioengineering models. It is no surprise to ethicists that this is seen as a troublesome bottleneck in the development and implementation of medical digital twins. In their discussion of the ethical implications of the engineering paradigm behind medical digital twins, Bruynseels et al. [32] suggest that clinicians have the potential to, at the very least, be caught up in current engineering practices and take these to a “different level” in medical contexts. The issue of adopting an engineering perspective opens the route to engineering actions being implemented in current healthcare and therapy. This, in effect, will impact on (1) whether life extension achieved through a digital twin is categorised as ‘therapy’ or ‘human enhancement’, and (2) our existing conception of ‘normality’ or ‘health’ [32]. There is obviously a danger here. In order to treat the patient, we may end up disembodying the digital twin—making it into something purely technical and letting human elements disappear via tweaks and refinements. Bruynseels et al. [32] raise the concern about how engineers can inscribe certain standards of normality into the digital twin, therefore enforcing a realisation of an ‘ideal’ human being, and likely to bring both moral damage to the clinician and physiological damage to the real patient (and society).

A third potential barrier to its implementation is the loss of control and autonomy. What if a clinician loses professional control over a digital twin that acts on behalf of the patient or their organ? [32,33,76]. An artificially intelligent digital twin implies a constant fulfilment of automated decisions on what is good or bad for the patient [32]. This calls into question: how much autonomy does the clinician have in making clinical decisions alongside automated decisions, recommendations or different types of outputs from the digital twin? Previous work has highlighted how AI systems have the potential to introduce over-reliance on automated decisions of AI systems which in the long term can lead to the phenomenon of deskilling [88,89]. A major issue related to overreliance in this context is how the value of existing patient autonomy is also likely to change in view of digital twin users having a strong dependency on the digital model [32]. Given the close link between digital twin and patient, a conflict of patient autonomy may arise if the digital twin’s decision on what is good or bad are prioritised over the patient, leading to a new form of “medical paternalism” [32]. 

In addition to clinical experts, the autonomy of digital twins may also result in the loss of control and autonomy for patients where a patient does not have control over their simulation that stands or acts on behalf of the individual or their organ [32,33,76,86]. The idea of letting a simulation act as a representation of a person’s organ or even their entire body forces us to think about how the digital twin has the capacity to shape the health of its patient, either for the patient’s benefit or the clinician’s own benefit. For Braun [76], this raises the problem of having “illegitimate representations” as a consequence of individuals not having control over their digital twin and not being aware that their current representation is actually not representative. This shift from representation to *illegitimate* representation allows the digital twin or its clinical users numerous opportunities to benefit directly from illegitimate forms of prediction or surveillance through learning more about their patient’s circumstances, including possible scenarios, and then acting on these scenarios. In particular, the role of assigning the representation to a scenario not usually accorded (i.e., where the simulation acts in the name of the patient), could lead to infringements on peoples’ personal freedoms in dealing with their own condition [76]. This raises monitoring and surveillance (including self-monitoring) concerns which were discussed earlier in the report. The great danger here is one of superficiality and of excluding or neglecting the embodied experiences of patients in their everyday life. Because of this potential risk we must ask the question: what if a clinician (or the digital twin) “modifies the manner of representation, transforms it into another form, interrupts or even ends it” [76]. This paradigm of shared probabilistic simulations is also intimately linked to questions of justice, for example, who should and must have access to simulated forms of representation? [76]. Furthermore, if digital twins are about providing clinical experts with real-time access to a patient’s representation, this predictive capacity is expected to ask bigger questions, involving more than their clinical knowledge, or as in the barrier presented here, crossing the divide between clinical practice and technical practice. 

### 3.7. Embedding and Adaptation over Time

#### Challenge to Scale Up

Table 7 is the seventh of the seven domains: embedding and adaptation over time (future outlook). It is believed that each member of society could eventually have a personalised digital twin for personalised medicine and treatment [32,83,84]. Currently, medical digital twin applications are at a very early stage of development, and there is substantial uncertainty on a time frame that helps to strategically accommodate the development and implementation process. This has led to some concern about medical digital twins being unrealistic because their complexity may take too long to produce meaningful results, or their organisation may struggle to continue developing the technology due to increasing costs which may derail the program [40,86]. There must be a monumental desire to keep the system running with hospitals and healthcare systems working together to make adaptations to improve its embedding. This may widen health inequality even more because the technology: (1) might not be accessible to everyone or not covered by some health insurers; (2) is likely to be implemented in predominantly North-Western, rich countries with access to research and development facilities; and (3) is at risk of being built on top of existing biases [80]. Other forms of potential injustice could occur from private financing of digital twin research and development, a problem exasperated if private sources want a return on their investment and drive the cost of the technology and its infrastructure up.

## 4. Discussion

In this article, we analysed academic publications and media publications authored by science and technology correspondents writing about digital twins in medicine, and employed the NASSS framework to identify the seven different domains the technology requires for implementation. Before listing these, it is worth noting that whilst there has undoubtedly been much promise in relatively constrained model development with retrospective data, there are no notable digital twins for personalised medicine. It should be clear from the findings that analysis is significantly weighted towards domain two (‘the technology’), compared to other studies using the NASSS framework (e.g., [23]). This is reflective of a more general truth that we are a long way away from any meaningful clinical implementation—that is, at the time of this publication no digital twins have been implemented for personalised medicine or in organ-based medicine, generally. The technology is still very much in its early development phase where considerable energy is being expended on collaborations within academia or industry or across both (for example, cross-sectoral collaboration). Much effort is being made to create cross-sectoral centres for doctoral training which focus on digital twin development, and considerable support is being provided to universities to set up the technology, train PhD students and postdoctoral researchers in its assembly, and support a collaborative approach to projects working on developing digital twin technologies.

In domain one (‘the condition’), *meeting a demand for preventing cardiovascular disease* with digital twins was identified as a key theme across the literature. In this domain, cardiovascular disease, as the number 1 cause of death globally, was seen to facilitate the development of digital twins in cardiovascular medicine, while the complexity of cardiovascular diseases among clinical experts was identified as a barrier. While digital twins were appreciated by clinical experts to help them tackle a spiralling problem, some experts were doubtful about the potential of such a system since cardiovascular disease had many risk factors (physiological, socio-economic, and socio-demographic) and often coexisted with comorbidities.

In domain two (‘the technology’), three key themes were identified: *technical functionality and reliability*, *technology features*, and *data*. Three facilitators were identified to support the development of tools for *technical functionality and reliability*: real-time synchronisation techniques, IoT, and cloud computing. Such facilitators certainly came with their own challenges; but it was clear that the biggest barrier to the implementation of digital twins for cardiovascular disease came in the form of hacking and viruses. Although hacking and virus risks are not new to technologies with ‘over the air’ Internet connection, digital twin developers need to consider how to address these problems with the utmost urgency given how medical devices (e.g., pacemakers) have already been subject to remote hacking or infection [90,91,92], implying that devices connected to networks are at risk of tampering. This barrier also indicates that implementing digital twins and sensors in hospitals or organisations connected to a network are at risk themselves of being hacked [93].

In terms of theme two: *technology features*, five facilitators were identified to support the implementation of digital twins: real-time simulation, real-time experimentation, real-time remote monitoring of a patient’s health, predictive maintenance, and virtual collaboration. Although the technical features mentioned above were easily imagined, we identified only two barriers. It was clear to us that the main barrier centred on the technoscientific expertise that clinical experts or staff needed to use the digital twin. Recognising exactly what type of training is required for clinicians to use digital twins is unknown. Strohm [85] has suggested that radiologists who use AI need to have some technical training in order to comprehend the outputs of the technology, despite computer science and programming knowledge not falling under the typical competencies of medical experts. In her research into the implementation of radiology AI applications, Strohm [85] reported that the radiologists who were proactive in learning the technical aspects of AI tended to be in the minority, yet for all their knowledge on the technical mechanisms behind AI applications they lacked knowledge in the form of guidelines or best practices mainly because “there is no scientific evidence available on these aspects”. If this is the case, then it is reasonable to expect that clinical experts or staff using digital twins will require training on how to use the system, but also how the system works.

The third theme: *patient data*, contains three facilitators (‘real-time data collection’, ‘data fusion techniques and multiscale data integration’, and ‘dedicated structured reading team’). However, each of these facilitators have limitations. In the case of real-time data collection and data fusion/integration techniques the technology is far away from being ready for application, and having a dedicated team to process the collection of patient data appears to be reserved for those who have substantial resources available to them, such as Siemens Healthineers. The theme of patient data, perhaps inevitably, carried with it many potential barriers to the privacy of patient data collected, processed, and managed from sensor technologies. The privacy of patient data is a persistent issue throughout the UK’s transition to a data-driven NHS [80,94]. Whilst independent Westminster Think Tank *Reform*—in collaboration with NHSX and input from regulators such as the Care Quality Commission (CQC), Information Commissioner’s Office (ICO), Health Research Authority (HRA) and Medical and Healthcare Products Regulatory Agency (MHRA)—have gone some way in providing rules that clarify access and use of NHS data by private sector companies for research or product and service development purposes, they have yet to fulfil the ideals of patient engagement and democratic participation. The contribution by NHSX [95] stresses how patients need to be engaged in what data are being used and by whom, but also the partnership and commercial relationships that exist [95]. If the NHS continues to pursue the commodification and selling of patient data as a way of building a healthier society and funding the NHS [94], questions still remain regarding the value of the data the NHS holds, and what counts as fair compensation [96]. As Nicola Perrin, then lead of the Understanding Patient Data initiative, commented in the House of Lords report [94], the public *‘do not like the idea of the NHS selling data, but they are even more concerned if companies are making a profit at the expense of both the NHS and patients’* [94]. Furthermore, as the production of new personal information based on AI and Big Data analytics becomes increasingly out of the individual’s control, the meaning and language around “ownership” of data becomes increasingly blurred [97]. For example, it has been reported that patient data from GP surgeries in the NHS have been sold to pharmaceutical companies in the United States for research, with drug giants paying the Department of Health and Social Care up to £330,000 each for anonymised data [98]. The question here is not so much about stopping the selling of data to research and private sectors to tackle serious healthcare problems, but to ask whether patients should be told that their data can be sold in the first place and to whose benefit [96]. Preserving the privacy of patient data functions just as much as a barrier as a facilitator in medical digital twin development and presents a challenge to research and development let alone implementation [86].

In domain three (‘value proposition’), our analysis identified two facilitators: that digital twins had the potential to be ‘cost effective’ and would lead to ‘personalised and improved treatments’ for patients, leading to the theme of *desirability*. However, this information tended to be somewhat speculative. The assumption was that using medical digital twins led to personalised diagnosis and treatment plans, and thereby streamlining the healthcare system’s priorities and costs and reducing healthcare expenditure [12]. However, these assumptions are based on the development and implementation of hospital or ‘strategic planning’ digital twins [63]. There is currently no evidence to suggest that the same will happen of medical digital twins being developed for personalised medicine given the complexity, heterogeneity, and changing conditions of human physiologies. Within the NASSS framework, the concept of value goes beyond simple financial measures, and compromises non-financial benefits, such as the trade-off between the potential benefits of using the technology and the work required to use it [15,85]. At the value level, taking into consideration the promises given in personalised medicine and using digital twins to customize patient care, it is currently impossible to know whether digital twins actually provide a clear business case for stakeholders. This is because efficacy or cost-effectiveness studies are unavailable or contested [32,36]. 

In domain four (‘intended adopters’), it was evident that the adopter system was complex. Given how most of the literature was based on the development of digital twins to tackle cardiovascular disease, it was recognised that the intended adopters of digital twin applications were predominantly specialist clinical staff in the cardiology context (such as cardiologists) who will directly use the twin (e.g., [69]). The lack of research of other direct users, but also a range of other professionals (e.g., nurses, general practitioners, technical staff) and organisational structures is, perhaps, related to a concentration of the efforts of digital twin developments in cardiovascular medicine. However, the use of sensors adds to the complexity of adoption and implementation processes because patients will be involved in the management and use of their sensors (as well as other IoT devices). While developer–clinical collaboration was seen as a key facilitator to adoption, our analysis of the literature suggests that clinician–patient collaboration is just as much a key factor to the clinical expert’s adoption of the digital twin. This type of collaboration points to the idea of equipping patients with sensors or other monitoring systems to help build their digital counterpart. While the breadth of the NASSS framework proved useful for the detection of five facilitators, we were only able to detect one example that might lead to the abandonment of sensor technologies. Apart from the barrier of sensor damage and failure caused (e.g., [59,77]), we were unable to identify other barriers to clinician–patient collaboration because this research does not yet exist. For this reason, we must briefly turn our attention to existing research on self-monitoring technologies which has gained a place of prominence in the medical sociology literature [99]. Existing research has noted patients’ everyday monitoring practices of blood pressure or weight as part of ‘healthy living’ [82,100,101]: this has opened up new questions and practices for those who ‘self-monitor’ and how these technologies work, not so much in terms of the technological nature of the technology or even the method per se, but how they work in the everyday life of the patient. These include more subtle issues concerning (1) the possibility of non-use and resistance to sensors; (2) the work required to insert sensors into domestic spaces and routines, and (3) the emotional meanings of these sensors and practices [82].

Despite this lack of research, the NASSS framework helps us suggest that patient empowerment may act as a key facilitator to the patient’s adoption of sensors in the digital twin context [36,55]. This finding echo’s existing research on self-monitoring technologies, which found that the ability to self-monitor and manage one’s health also brings into play the potential for patient empowerment [81]. Patient empowerment is said to be defined by the information and abilities a patient gains in order to actively partner with healthcare professionals regarding decisions about their health [102]. Empowerment can be described as a set of competencies in which patients improve their health literacy (e.g., [103]), control [104], participation [105], and communications capacities [106]. While empowerment is widely regarded as having a positive impact on the patient, it has also opened up two negative matters of patient adaptation. First, it opens up a more insidious aspect of this type of relationship; that people are led to believe they are being empowered when in fact they are being exploited [82]. If this is the case, digital twin users need to open a dialogue with patients and reassure them of the system’s security [107], and how patient-generated data will not be exploited or sold to for profit-companies. If this topic emerges patients must be told and compensation mechanisms must be determined [32,33]. 

Second, it opens up the problem of whether patients can live an independent and satisfying life [37,108]. Tadas and Coyle [37], in their literature review assessing the barriers and facilitators of different digital interventions in cardiac rehabilitation and self-management (e.g., mobile phones, sensors, and telehealth), for instance, highlighted how patients reported an “increased burden” when using the technology in their everyday life. This burden is partly directed toward healthcare professionals asking patients to add their device on top of what they already use (e.g., weight scales and blood pressure cuffs), leading patients to become side-tracked and to not use it every day [37]. In particular, patients in older age groups lacked interest in using monitoring technologies, citing forgetting to use it, losing it, and inconvenience as a barrier to self-management and use of the technology [109]. Furthermore, practices of self-monitoring have highlighted the issue of patient anxiety because of home monitoring and how efforts must be put in place in order to “mitigate” patient anxiety [101]. These issues strengthen the call for sensors that are either minimally invasive or miniaturised and require little engagement from the patient. By bringing in the existing research on self-monitoring practices, we have been able to raise potential barriers that may impede both the development and implementation of digital twins. In a multi-adopter world, this calls for future research which embeds clinician–patient collaboration early on in the development process for introducing sensor technologies to patients.

In domain five (‘the organisation’), there was a total absence of an organisation’s readiness for this technology, and the extent of work needed to implement changes in hospitals or the wider healthcare system. Research by Fuller et al. [40] highlighted the financial and business challenges associated with medical digital twin implementation, including the cost of installing and running a high-performance IT infrastructure that can support the performance of algorithms, ensuring a system is in place to ensure data are of the highest quality, privacy, and security, and an IoT system that enables seamless integration of sensors. Because of these challenges, three barriers were identified (‘existing IT infrastructure’, ‘high cost’, and ‘recruitment of data science team’). Despite these barriers, we could not identify how a digital twin for cardiovascular disease would fit with existing organisational routines. Previous research using the NASSS framework identifies how new medical technologies create opportunities for developing new routines and care pathways, but also have the potential to disrupt existing patterns of clinical work in ways that complicate adoption [23,25,38,85,110]. Following this limitation, there needs to be a new wave of research and thinking on activities and organisational factors for digital twin implementation that take into account the integration of medical digital twins with existing routines and the extent to which it aligns or clashes with existing organisational and interorganisational routines.

In domain six (‘the wider system’), there was a total absence of support from the wider system given how no digital twins for personalised medicine had been tested and deployed in society. Whilst there is a clear movement for the development of AI enabled solutions in personalised medicine (e.g., [45,80,94,95]), there has been very little mention of digital twins in the regulatory and professional bodies discourse. Although there is a rise in medical AI systems being implemented worldwide, rigorous regulation on implementation issues for diagnosis and treatment applications is still lacking [111]. Whilst efforts have been made in the past year to regulate medical AI, there is still a lack of empirical evidence on implementation factors of AI technologies in clinical practice (e.g., effects on diagnosis and patient–clinician relationships), and failing to reflect the breadth of these practices and scenarios may lead to barriers of implementation [111]. Although studies had started to ask questions related to governance (e.g., [6,11,14] regulation and legal aspects were not the sole focus of these articles and tended to be in the form of editorials which lacked detail (i.e., [41,42]). Because of this, no steps had been taken by regulators to standardize the production of evidence or make a significant impact on policy making and risk management. Interestingly, all of these studies included criticisms of how the existing regulatory landscape for medical devices was a barrier for medical digital twins, ranging from “being populated by too many regulations” [86] to the “cost, timeline and perceived rigor” of the approval process [14]. Nonetheless, the notion that computational modelling and simulation tools could be used to aid device evaluation and reduce costs was seen as a potential facilitator in the overall implementation process. Among the most important barriers in this domain are the potential professional, cultural, and ethical issues related to human enhancement and loss of control and autonomy (e.g., [32,33,76,86]. While ethical risks such as discrimination and data privacy are nothing new in AI-driven medicine, what is new are the forms of AI-driven simulations that take on forms of representation and act on behalf of a physical person. This shift in machine intelligence is likely to create decisive ethical concerns over the control of the simulation which may, “rise up against its biological counterpart” [32] and make decisions on what is good or bad for the patient, and thus compromise the critical reasoning, medical judgements, and professional autonomy. Thus, ethical barriers for digital twins appear to qualify as complex. This complexity can be related to the fact that ethical barriers combine three elements: concerns of human enhancement, loss of control and autonomy, and privacy. To varying degrees, these three elements also apply to other AI applications in medicine, yet we may see new issues around discrimination or equality emerging [14].

In domain seven (‘embedding and adaptation over time), there was a total absence of site-specific iteration and embedding of a digital twin for cardiovascular disease given how no digital twins for personalised medicine have been implemented. Future studies should consider undertaking an exploration of such factors to assist in providing a more comprehensive understanding of how digital twin implementation evolves over time in different clinical settings, especially if these systems are being developed for primary care and secondary care settings. Strohm [85] argues that further research is important in this domain due to the non-linear nature of the implementation process and is especially important for AI-driven systems because they create a consistent demand for adaptation over time. Fuller et al. [40] and Popa et al. [86] remind us that the possibility of creating a dynamic replica of a human organ or body requires a dynamic system which continuously adapts to operational changes based on updates from model and sensor information. The latter study is of particular interest here as it draws attention to digital twins increasing rather than reducing health inequalities due to the high infrastructural costs of embedding a system that requires significant customisation and ongoing adaptation. 

## 5. Conclusions

This study is a NASSS-informed [15] document analysis of peer-reviewed academic research and media articles on barriers and facilitators of digital twin implementation in cardiovascular medicine. The identified barriers and facilitators, as well as complexity levels of different domains of the innovation at such an early stage, gives us the opportunity to learn what developers and potential adopters of such digital twins require for successful implementation. The presentation of a high number of facilitators (*n* = 11) and barriers (*n* = 9) in domain 2 (‘the technology’) highlights the vast number of journal articles in this domain and a high degree of complexity in the development of digital twins for cardiovascular disease. Facilitators ranged from promises of ‘real-time synchronisation between virtual and real entities’ to ‘real-time simulation’, whereas barriers ranged from ‘hacking and viruses’ to ‘specialised skills required’. However, this intense research focus on medical digital twin development has meant that the following domains are limited to speculative commentaries, mostly from media coverage of medical digital twins. This is particularly problematic for domain 4 (‘the intended adopters), where there is a total absence of the characteristics and behaviour of adopters, and the nature and extent of different interactions and influences on adoption decisions. Popa et al.’s [86] study stands out in this regard, although it is in fact a collection of views from potential adopters projecting their future view on the technology which means it lacks any present ability to report on adopters’ real use of twins or sensors.

Fascinatingly, this may be the reason why domain 5 onwards reveals a shift towards there being more barriers than facilitators, presumably because these domains lack evidence on the realities of implementing this technology in clinical settings and the effects of these technologies on organisations, ethics, regulation, and finances. For example, domain 6 outlines one facilitator (‘potential of using modelling and simulation tools to help produce standards and guidelines’) but four barriers (‘lack of regulation for clinical practice’, ‘regulatory authorities’, ‘concerns of human enhancement’, and ‘loss of control and autonomy’). 

Our findings suggest, therefore that most of the research being conducted is primarily focused on the development of medical digital twins, clearly detached from social, organisational, and legislative/regulatory aspects. Although there is clear motivation to prevent cardiovascular disease and the determination to create digital twins to improve the health of society by identifying and evaluating techniques to prevent disease, this development is occurring with very limited empirical work on the social, political, ethical, economic, and regulatory challenges that implementation would likely bring into being. Current insights on these matters are drawn from articles which promise or speculate about the future. This means that there is very limited empirical work that provides evidence on the development or implementation process of the technology, with Popa et al. [86] being the only exception. From this perspective, our use of NASSS as an analytical tool helps to catalyse empirical work in social sciences and humanities which not only bridges the gap between the technical and the social but also ventures into fundamental questions of Responsible Innovation [24]. Finally, we note that our classification of each domain tended to be ‘complex’, meaning that the technology or innovation is less likely to be successfully adopted, scaled up, spread, and sustained [15].

Our research, then, indicates the importance of building a bridge between the complex technical and social aspects of those seeking to design, develop, implement, scale up, spread, and sustain digital twins in cardiovascular medicine. In order to make digital twins in medicine possible, we call for greater understanding in how these systems are developed and implemented, as well as how social scientists can work with developers and society in designing responsible digital twins. Doing so, might bring balance to facilitators and barriers across the domains and as a consequence may enhance its chance for adoption and implementation.

## Figures and Tables

**Figure 1 sensors-23-06333-f001:**
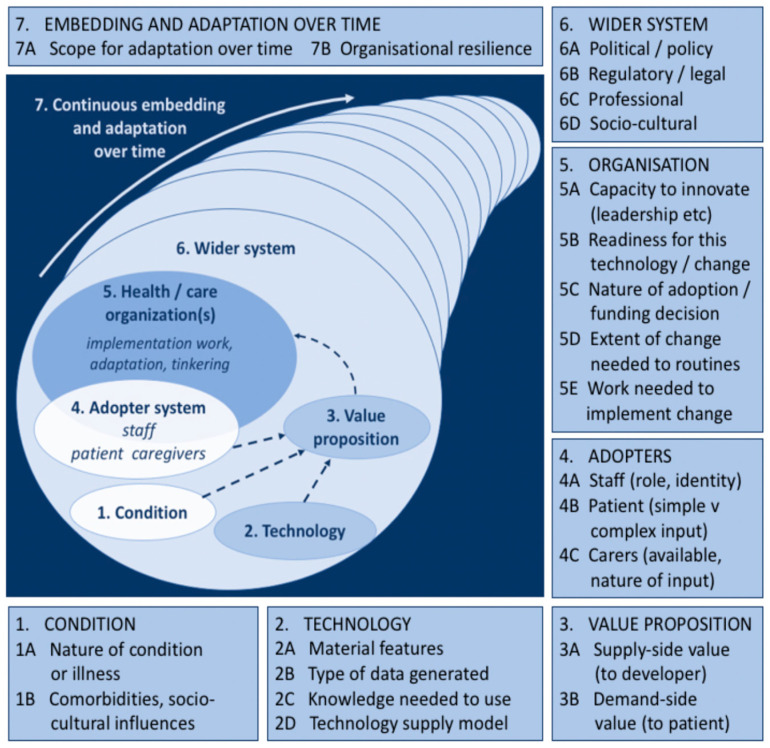
The NASSS framework by Greenhalgh et al. [15].

**Table 1 sensors-23-06333-t001:** Domain 1: The Condition.

Complexity Level	Theme	Facilitator	Barrier
Complicated	A demand for preventing CVD	CVD is the number 1 cause of death globally [4,11,14,26,45,46,47,48,49]	CVD is a complex disease and influenced by physiological, socio-economic, and socio-demographic factors [1,11,14,39,47,48,49]

**Table 3 sensors-23-06333-t003:** Domain 3. Value Proposition (for digital twin developers and users).

Complexity Level	Theme	Facilitator	Barrier
Complex	Desirability	Managing cardiovascular disease is costly [6,14,26,34,35,36,58]	
		Personalised and improved treatments [1,6,11,31,34,35,36,42,58,60]	

**Table 5 sensors-23-06333-t005:** Domain 5. The organisation.

Complexity Level	Theme	Facilitator	Barrier
Complex	Changes to existing system		High cost [14,39,40,57]
			Existing IT Infrastructure [14,39,40] Recruitment of data science reading team [75]

**Table 6 sensors-23-06333-t006:** Domain 6. The wider system.

Complexity Level	Theme	Facilitator	Barrier
Complex	Societal development	The potential of using computational modelling and simulation tools to help produce standards and guidelines [11,14,57]	Lack of regulation for clinical practice [14,41,42,57] Regulatory authorities [1,6,8,11,14,32,33,40,41,42,64,86,87].
			Concerns of human enhancement [32] Loss of control and autonomy [32,33,76,86]

**Table 7 sensors-23-06333-t007:** Domain 7. Embedding and adaptation over time (Future outlook).

Complexity Level	Theme	Facilitator	Barrier
Complex	Challenge to scale up	Each member of society could eventually have a personalised digital twin for personalised medicine and treatment [32,83,84]	Fear about digital twins being unrealistic [40,86] High economic cost [40,80] Widening health inequality and injustice [40,84]

## Data Availability

Not applicable.

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
