# Peer review of "Using the Non-Adoption, Abandonment, Scale-Up, Spread, and Sustainability (NASSS) Framework to Identify Barriers and Facilitators for the Implementation of Digital Twins in Cardiovascular Medicine"

_sensors, 2023, doi:10.3390/s23146333_

Round 1

Reviewer 1 Report

This paper presents a high-level and "prospective" analysis of Digital Twin as a technology for adoption in a particular application area (cardiovascular medicine). The novelty is clear;  the methodology is sound; and the narrative easy to follow. In my view, it could be publishable as is. However, I believe the following editorial suggestions would make the paper more readable for the typical reader by making the most read parts more self-contained:

(1) The first part of the title is eye catching but could be misleading (as it was to me initially). It's not about (technical) implementation of DT per se. The second part of the title reflects the real content. It would be more to the point if the title was "Using the NASSS Framework to Identify Barriers and Facilitators to Implementation of Digital Twin in Cardiovascular Medicine".

(2) The novelty in the rather long Introduction section isn't stated as explicitly and clearly as in Conclusion. 

(3) It might be worth summarising the main points in Conclusion. 

Reviewer 2 Report

In the first place, they present a very extensive work with a high number of data and details of the reviewed articles.

Although this is very good, what also originates is a very long manuscript where a lot of data is repeated, which makes the article a bit dense for reading.

In summary they talk about digital twins, but instead of focusing on cardiovascular twins, the authors could talk about digital twins in general since they focus on cardiovascular twins later on. 

I would suggest authors create a table summarizing all the articles mentioned so that the reader can quickly view all the articles in the review.

The current structure presents a high density that complicates the agile reading of the text.

A fact that is very weak is the writing of conclusions.

After everything described and the points exposed, with the different domains, cases, problems of the use of computer data, possible hacking, etc., the authors do not express themselves in an exhibition of conclusions with possible existing solutions.

The wording of the conclusions should be more explanatory for the reader, including solutions or possible solutions to some of the problems described throughout the text.

Author Response

Please see attachment 2

Reviewer 3 Report

A manuscript with a very interesting and important title. The abstract needs some refinement. I recommend that the authors provide the purpose of the work, what methodology was used and when, and what results of the work were obtained. I recommend that the information about barriers be removed at this point. The work lacks information on what digital twins are, the origins of the issue, and the purpose of their creation. The authors in the introduction use the abbreviation NASSS, but do not explain what it stands for. There is also a lack of development and justification for the use of NASSS at work. Conclusions need to be refined in relation to the purpose of the work, the results achieved, and achievements in the literature on the subject. The strength of the work is the title and methodology. Weakness is an abstract and too short conclusion. The article requires linguistic and stylistic corrections. It also recommends adding literature from 2023. I recommend removing the 1995 literature from the paper. I recommend inserting the following item:

https://doi.org/10.1016/j.resourpol.2023.103345

The article requires linguistic and stylistic corrections.

Author Response

Please see attachment 3

Round 2

Reviewer 3 Report

The manuscript has not been adapted to the recommendation. The manuscript contains numerous grammatical errors. For this reason, I cannot accept it in its present form.

The manuscript contains numerous grammatical errors. For this reason, I cannot accept it in its present form.
